# Calorimetric Measurements of Biological Interactions and Their Relationships to Finite Time Thermodynamics Parameters

**DOI:** 10.3390/e24040561

**Published:** 2022-04-16

**Authors:** Yuwei Zhang, Gregory J. Kowalski

**Affiliations:** Department of Mechanical and Industrial Engineering, Northeastern University, Boston, MA 02115, USA; zhang.yuwei@northeastern.edu

**Keywords:** calorimetry, entropy flow, entropy production, biological communities, reacting systems

## Abstract

A description and examination of the potential for calorimetry for use in exploring the entropy flows in biological and or reacting systems is presented. A calorimeter operation background is provided, and two case studies are investigated using a transient numerical simulation. The first case describes a single cell calorimeter containing a single phase material excited by heat generation source function such as joule heating. The second case is a reacting system. The basic observation parameter, the temperature, cannot be used to separate the entropy property changes and the rate of entropy production in the second case. The calculated transient response can be further analyzed to determine the equilibrium constant once the reaction equation and stoichiometric constants are specified which allows entropy property changes and the rate of entropy production to be determined. In a biological community, the equivalent of the reaction equation and a definition of an equilibrium constant are not available for all systems. The results for the two cases illustrate that using calorimetry measurements to identify the entropy flows in biological community activities requires further work to establish a framework similar to that chemical reacting systems that are based on an equilibrium type parameter.

## 1. Introduction

Calorimetry is utilized to measure the thermodynamic properties such as enthalpy, entropy, and Gibbs free energy in biological and chemical reacting systems. While the energy/heat released during these processes is important, the entropic changes which reflect the irreversibility and efficiencies are equally important in the understanding of them. Calculating the heat released from one single microbe has confused biologists for centuries. Most marine microbe studies measure the heat dissipation on a community level since the isolation of individual types of microorganisms is not required. In a drop (one millimeter) of seawater, there are approximately 0.5 million microbes and 10 million viruses. To describe the distribution of individual bacteria is time-consuming and biased in their culture practice. Djamali et al. [1] estimated that the heat release per marine bacteria is 50 nJ after concentrating the population by filtration and backwashing. Roach et al. [2] collected the rate of heat-released data by a TAM III calorimetry from various microbe groups and found it to be approximately 25 J/s per gram of seawater from an aquarium system. Precise thermodynamic data is demanded since the measurement of microbe activities is on a nanotechnology scale, much smaller than observed in pharmaceutical reactions [3,4]. The output from the commercial calorimeters is usually reported as power, μW, or often using units of $\mu \mathrm{cal}\cdot s^{-1}$ or $\mu J\cdot s^{-1}$ as a function of time [3] and converting to energy units, J. The observed output is the change in the power used to control the sample and reference cell maintaining the same temperature. Maintaining two cells in a calorimeter at the same temperature is intended to have the same heat exchange rate between them and the surrounding bath in order to observe energy released by the biological process. The issue is how to determine the entropy changes in biological processes and the entropy flows that occur in these devices.

The Gibbs free energy is considered an important parameter to understanding the thermodynamic information on living organisms and the driving force for chemical processes of metabolism activities. The change in Gibbs free energy can be estimated for known entropy of the biomass or with Roel’s correlation [5]. However, current calorimetric studies have not developed a way to directly measure the Gibbs free energy from microbial experiments, nor to measure the entropy change in the reactions occurring in the community [5,6,7]. The Gibbs energy dissipation is related to entropy production exporting from the cell into the environment. The change in entropy includes entropy flow, which is caused by the exchange of entropy in the system with its surroundings; an entropy property change, the entropy exchange between different states of the system and is related to the mass, heat specific, temperature, and pressure. The entropy production is related to the irreversibility of the process [8]. Having an understanding of the entropy flows and the Gibbs free energy change as a biological community interacts with its surroundings provides a more in-depth knowledge of the efficiency of these systems and the tradeoff between switching to maximum power (energy rate) modes vs. energy efficiency modes.

The actual spontaneous processes in biological systems are irreversible and will have an entropy production that adds to the observed heat dissipation in calorimetric measurements. There is a tradeoff between the efficiency of biological processes and the processes rate, with the maximum rate of operation accompanying a 50% loss of the yield [9]. The biological processes are driven by the flow of exergy through the informed pathways rather than the production of entropy. The concept of dissipative biological structure describes these living systems from microorganisms to ecosystems by the dynamics of a far-from-equilibrium system [9]. In contrast, the calorimeter measurements are closed systems after introducing nutrients or competing phages or viruses to the sample. In both cases, it is desired to have measurements that would lead to identifying the entropy property changes and entropy production parameters.

This paper aims to examine what the calorimetric measurements provide and to introduce potential novel methods that determine the equilibrium constant from calorimetry measurement and use it with the second law of thermodynamics to relate it to the Gibbs free energy change. We demonstrate that calorimetric experiments by themselves do not provide the information to isolate the entropy production term, the irreversibility, in the observed reaction. In the case of the pharmaceutical industry, calorimetry plays an important role but the additional analysis which includes an assumption of the reaction type is required to obtain the necessary entropy information. While this step is part of the commercial calorimetry package, its use toward identifying the irreversibility and potential of a drug–protein interaction is clearly understood in this process. In terms of biological processes, the role of calorimetry is increasing. as is the thermodynamic and entropic interpretation of these processes. The manuscript identifies the missing steps in linking calorimetry and other known means related to growth rate formulations, degree of reduction or reactions that would allow this experiment tool to provide entropic information.The paper focuses on describing the thermodynamic and heat transfer physics in the calorimeter device and their relationship with observed energy transfer and entropy flows. These relationships are fundamental and apply to the different time scales seen in drug discovery processes and biological systems. The method of thermodynamic properties measured from the isothermal titration calorimeter will be discussed and extended to a single injection type of experiment, which is related to that biological sample. In the following section, the calorimeter background with two cases will be provided to illustrate how entropy production and entropy properties become convoluted during the measurements.

## 2. Calorimetry

### 2.1. Background 

Two different configurations of a calorimeter that could be used to measure the energy released by microbes, phages, and virus, their interactions, or drug-related chemical reactions are shown in Figure 1. 

In Figure 1A, a typical isothermal titration calorimeter (ITC) or two cell configuration that has been involved in the previously cited papers [3] is illustrated. This configuration is usually operated at a constant temperature and in biological situations is not operated in a titration mode [2,10], but the operating principle is similar to that described below. In this device the two cells, reference and sample cells, are similarly constructed so that the heat transfer between them and the surroundings can be considered to be the same if they are maintained at the same temperature using electric heaters. The heaters are controlled by varying the power to them to maintain the same temperature in each cell where the respective cell temperatures are measured by either thermopiles or other accurate devices. 

The rate of energy released by the observed reaction in the sample cell; for example, a microbe community with a food source, will cause the temperature of the sample cell to change unless the power to it is adjusted to maintain a constant temperature. The observed variable in these measurements is the difference between the reference and sample power, Δ*P*, which is stated to be equal to the rate of energy released during this process. The rate of energy released ultimately is transferred as heat to the surrounding large thermal reservoir. The energy released associated with the process under the assumption of constant temperature in each cell and that the cells are similarly constructed lead to a simple relationship, Equation (1).
(1)$$\Delta P=Observed\;Change\;In\;Power=P_{REF}-P_{SAM}={\dot{E}}_{REL}$$
where *P_REF_* and *P_SAM_* are the power to the reference and sample cell, respectively. ${\dot{E}}_{REL}$ equals the rate of energy released by the reactions occurring in the sample cell. The energy balance on the sample cell provides additional insights into these assumptions and the mechanisms that occur during the reaction, Equation (2).
(2)$$P_{SAM}+{\dot{E}}_{REL}=\frac{dU_{SAM}}{dt}+q_{L,s}$$

In Equation (2) $U_{SAM}$ is the internal energy change in the materials in the sample cell, *t* is time and $q_{L,s}$ is the rate of heat transfer between the sample cell and the surrounding bath. While in most of these devices, there is an injection to the test sample to add food or duplicate injections requiring a transient form of the energy balance to simulate the process. It should be noted that the rate of change of the internal energy depends on the temperature change in the sample, as well as the changes in its mass. In these calorimeters the temperatures of the reference and sample cells are controlled to be equal and set to a constant value. For these conditions, the derivative of the internal energy term will be zero; if the mass of the sample cell remains constant during the measurement, at steady state conditions, or when a balance of the interacting group of various species is reached, the transient term on the RHS of Equation (2) will approach zero. Assuming that the working fluid in the device is an incompressible material yields Equation (5). The time dependence of the mass in the sample cell is related to the mass flow rate of the injection, Equation (3) and the expanded form of the internal energy, Equation (4) is included in the transient energy balance, Equation (5), to emphasize the above assumptions.
(3)$$m_{sam}={\dot{m}}_{inj}t$$
(4)$$U_{SAM}=m_{sam}c_{s}\left(T_{S}-T_{R}\right)$$
(5)$$P_{SAM}+{\dot{E}}_{REL}=\frac{d\left(m_{sam}c_{s}\left(T_{S}-T_{R}\right)\right)}{dt}+q_{L,s}\;\;\;\;$$

In Equation (4), *m_SAM_* and *c_s_* equal the mass of material and the specific heat of the material in the sample cell, *T_s_* is the temperature of the sample cell at any time and *T_R_* is the reference temperature for the internal energy. The mass of material, $m_{sam}$, in a typical titration experiment is related to the injection rate, concentration, and injecting time. $\dot{m_{inj}}$ is the injection of the syringe in Equation (4). A similar energy balance can be written for the reference cell where there is no rate of energy released term. Since the temperature of both cells is controlled to be equal in the theory of two-cells ITC and they are geometrically similar, the heat loss from both cells will be the same at any time. The internal energy balance in the reference cell is constant as its mass does not vary with time. Since the temperature of each cell is controlled to a constant value, the temperature time derivatives are also zero. The implications of these assumptions will be discussed later. Combining these energy balances of sample and reference cell under these conditions yields the following relationship:(6)$${\dot{E}}_{REL}=c_{s}\left(T_{S}-T_{R}\right)\frac{d\left(m_{sam}\right)}{dt}+\left(P_{REF}-P_{SAM}\right)$$

The derivative term in Equation (6) was left to illustrate that even if the temperatures are controlled to be equal and constant, a part of the internal energy change will remain if the mass changes. Equation (2) assumes that the preceding condition is satisfied. If it is not, then the result summarized in Equation (2) is not valid.

Commercial ITC/sample and reference cell calorimeters are designed and manufactured to satisfy these conditions for chemical reaction processes. When these calorimeters are used to measure the rate of energy release from the microbe community, as in [1,2,10], the effects of injecting fluids at a different temperature need to be taken into account. As that of the cells, the effects include the changes in mass from gas exchange with the surroundings, separating the growth of the microbes, significant changes in the metabolic rate based on the community dynamics and potential interactions between the microbes and phages or viruses. If sufficient time is taken one can reach a near steady state condition, which avoids the initial injection region and the growth of the community phases. In standard ITC practice, the power difference which equals the energy release is integrated over the time of reaction to obtain the thermodynamic property of the enthalpy of reaction, Δ*H*, and is usually expressed per number of microbes or mole of injectant. In the case of the microbial community, this change in power is related to energy released by the microbes and has also been reported [1,2,10]. The heat transfer rate from each cell in these types of measurements is not usually reported, although the difference in the power between the reference and sample as a function of time is as reported in [2].

To understand what is measured by the change in power, the single-cell calorimeter as shown in Figure 1B is analyzed and it is essentially one of the cells from the ITC/ sample and reference cell device. In this case, the energy released, ${\dot{E}}_{REL}$, is determined from the observed temperature response, the heat input, *P_sam_*, if any, and the heat transfer characteristics of the cell, $q_{L,s}$. A calibration relation is required to determine these heat transfer characteristics. The energy balance on this device is similar to that of Equation (2) and can be solved for energy released, ${\dot{E}}_{REL}$. The integration of the energy release term over the time of the process can be used to determine the enthalpy of reaction as stated above.
(7)$${\dot{E}}_{REL}=\frac{dU}{dt}+q_{L,s}-P_{sam}$$

If the energy released by the community or the enthalpy of reaction was the only variable of interest and not the entropy flows in the system, the above experimental protocols would be sufficient [1,2]. In this case, the internal energy of the community is assumed to be a function of the temperature and mass of the fluid containing the microbes and other biological materials. Additional experiments to determine these parameters can be easily designed. The heat transfer from the cell is described in Equation (8) by the temperature difference between it and the surroundings and the overall heat transfer coefficient, (*u_HT_A*).
(8)$$q_{L,s}=\left(u_{HT}A\right)\left(T-T_{\infty }\right)$$

What is not obvious in both Equations (2) and (7) is that the heat transfer from the cell also would transport any irreversibility associated with the processes being observed which are convoluted with the heat flowrate from the sample cell. The details of this process are discussed in Section 2.1. The steady-state results stated in Equation (1) do not provide information on the irreversibility directly when a reaction is occurring and only the energy leaving the sample cell can be used to determine the rate of entropy flow from the sample cell. Other means are needed to provide information on the irreversibility or process efficiencies. For example, in (ITC) experiments, one performs the experiments sequentially to observe the reaction as the concentrations change in a prescribed manner in order to develop a relationship between the extents of the reaction and to determine the reaction’s equilibrium constant, *k_D_*. The equilibrium constant allows the introduction of second law parameters such as the Gibbs free energy change of the reaction, which then allows the entropy of the system during the reaction to be determined. A second law analysis of the calorimeter is then used to calculate the entropy production rate, which is related to the irreversibility of the reaction (Section 2.3). Procedures and hypotheses concerning the relationships for energy stored in growth, growth rate, and generation times would need to be applied to possibly formulate entropy statements for these processes, similar to that carried out in the ITC analysis [10].

### 2.2. Case 1: Spatially Uniform System Heating without Reaction 

Consider a simple system such as the single-cell, Figure 1B, that corresponds to an incompressible material, water, that is excited using microwave heating. This problem, while simple, provides the base solution to illustrate the determination of the rate of energy released, entropy flow, entropy property changes and the rate of entropy production from a calorimeter device. Assume that the single cell is a closed system, there is no expansion work nor mass exchange with the surroundings. A steady-state heat source, $q_{g}$ that represents the microwave or Joule heating per volume is active in the closed system, $q_{g}$, and there is heat exchange, $q_{L,s}\;$with the surroundings through convection (Equation (9)). The internal energy, *u*, is determined from the equation of state for the material in the cell. A is the surface area for heat transfer of the closed system of the test cell and the temperature of the surroundings is *T*_∞_. The energy balance is developed as in Equation (5).
(9)$$P_{SAM}+q_{g}\;V=\frac{d\left(m_{sam}c_{s}T\right)}{dt}+q_{L,s}$$

The initial conditions for this problem are:
*Time* = *t* = 0 *T*(0) = *T*_∞_(10)
(11)$$\mathrm{For}\;t\;≧\;0,\;\dot{q_{g}}=\mathrm{constant}=10^{6}\;W/m^{3}$$

The material in the single-cell system is water with constant properties of specific heat, *c_S_* = 4180 kJ/(kg K), thermal conductivity = 0.6 W/(m K), and density of 997 kg/m^3^. *P_SAM_* is set to zero in this simulation. The volume of the liquid is fixed at 80 µL and the (*u_HT_A*) product is 0.0356 W/K. The value of the power into the cell is zero in this experiment. Equation (9) was solved numerically using the code developed by Modaresifar and Kowalski [11] for reactingmixtures in microchambers, which simulates an injection experiment and analyzes its thermal process in a microscale calorimeter. This is a straightforward problem that is easily solved. This code was found to spatially converge to an accurate solution with three nodes for this small volume and uniform heat generation rate. It predicts the temperature time response, as shown in Figure 2, until the steady state is reached, approximately *t_f_* = 130 s. In Figure 2 the temperatures of the three spatial nodes are shown. They are not distinguishable from one another, which confirms that the numerical simulation satisfies the uniformly spatial assumption. Tests at a higher number of nodes, providing more precise resolution in the simulation, confirm that the solution has converged.

This case provides a baseline solution of what is measured in a calorimetric type of device and allows one to investigate the entropy flow from the sample cell, the rate of change of the entropy property of the material within it, and the rate of entropy production for this process involving a single material phase with known properties. To illustrate the above statement, one must view Figure 2 not as an end result of a simulation, but as the observed variable in the calorimetric device. As described in the background section, what one observes in the calorimeter is either the temperature response or the result of a small change in temperature that alters the power into the cell. For this discussion, Figure 2 is not the outcome, but the starting point to determine the heat generation term, *q_g_*, in Equation (9). This calculation is accomplished by integrating Equation (9) with respect to time under the assumption that *q_g_* is constant. In this step, note that one is specifying the function form of the internal heat generation, a constant, that is occurring in the sample, i.e., a mechanism. The integrated form of Equation (9) is:(12)$$\int_{0}^{t_{f}}q_{g}\left(V\right)dt=q_{g}\left(Vt_{f}\right)=\left(m_{sam}c_{s}\right)\left[T\left(t_{f}\right)-T\left(0\right)\right]+\int_{0}^{t_{f}}q_{L,S}dt$$

The heat generation term, the LHS of Equation (12), is usually the primary measurement sought, since it can be related to the energy released by the microbe community or a chemical reaction. Further analysis is performed to determine the irreversibility associated with this heating process, using relationships with the entropy flow.

The heat loss, *q_L,s_*, in Equation (12) is the heat transfer rate and is summarized in Figure 3 as a function of time using Equation (10). As expected, the heat transfer rate starts at zero and increases to its steady value. The difference between the heat transfer rate and the rate of energy released from the heat generation term is the internal energy storage rate, the first term on the RHS of Equation (12).

The flow of entropy from the sample cell is determined from the calculated heat flow and the temperature of the sample cell. This relationship is summarized in Figure 4. The rate of entropy flow out of the sample cell can be determined from the calorimetric measurements of the temperature and heat transfer information, as demonstrated below. The entropy flow out follows the behavior of the heat transfer rate as expected.
(13)$$S_{out}=rate\;of\;entropy\;flow\;out=\frac{q_{L,s}}{T}$$

The entropy property change is determined by the temperature, as observed in Figure 2, and the Gibbs equation. Water at moderate temperatures and pressures is used in this simulation and is treated as an incompressible material
(14)$$\frac{dS_{prop}}{dt}=m_{sam}c_{s}\frac{dT}{T}+\frac{P}{T}dv=m_{sam}c_{s}\frac{dT}{T}$$

The rate of entropy property changes at each observed time, Figure 5. This simulation case can be determined because one is using a pure material with a known equation of state. The temperature terms can be calculated for this case from the calorimetric observations. One feature of Figure 5 is the rapid increase in the rate of entropy property changes at the start of the process and then a steady decline until it reaches zero. During this response, the rate of entropy flow from the sample cell is increasing to its steady-state value, Figure 4.

The preceding calculations and graphs illustrate the different entropy components that are obtained from calorimetric measurements. The entropy balance is used to determine the rate of entropy production, which is related to the irreversibility observed for the constant heating process in this case.
(15)$$\dot{\sigma }=\frac{dS_{prop}}{dt}+\frac{d}{dt}{\dot{S}}_{out}$$

The rate of entropy production as a function of time is determined from the previously calculated values using Equation (15) and is summarized in Figure 6. One interesting feature is that the rate of entropy production increases rapidly and then slightly decreases, reaching a near-constant value, similar to that of the entropy flow out of the system.

For the heat generation process in this case, the rate of entropy production has a direct link to the irreversibility of this process. If this were a microbe community with a constant energy release rate, the results in Figure 6 would be a measure of the irreversibility and possibly would be related to the efficiency with which the community is using its resources. Such a case of near-constant energy release rates has been reported for *E. coli* during the stationary phase [10].

The above case is simplistic and a direct application of thermodynamics to a calorimetric type device. The results are expected and are presented here to establish; 1. what information could be obtained from a calorimetric type device and, 2. an approach to provide a baseline measure of the irreversibility. It neglects the complications of inter species competition for food sources, different phases such as exponential growth or declining, as well as changes in the immediate surroundings that one species may cause. These complications go directly to the assumption that the property relationships for calculating the entropy property solely on the observed temperature in an active community are possible. Without these key relationships, other means are needed to describe the interactions. Some of these complications are introduced in the second case, that of a sample cell containing a reacting mixture or interacting group of various species.

### 2.3. Case 2: Spatially Uniform System with Reaction Limited by an Equilibrium Constant

Case 2 considers the same sample cell as shown in Figure 1B and that was discussed in detail for case 1 for the constant heat generation case. In case 2, the sample cell is initially filled with two reacting solutions, where these solutions could reflect chemical reactions or two interacting species; for example, *E. coli* and phages or ligands. From an experimental viewpoint, the same observed property, the temperature, is measured. The reacting solutions will normally be low concentrations of the reacting species in a buffer or low concentrations of microbes in water that contains a food source. From a calorimetric viewpoint, the key rates of energy exchanges are the rate of change in internal energy of the buffer/water and the rate of heat transfer from the sample cell. The driving thermodynamic potential is the reaction that will be characterized in terms of reaction equation, equilibrium constant, growth rate constant, degree of reduction, or equivalent, and the rate of reaction. The reacting compound case will be used as a starting point to illustrate these relationships and what information we can obtain from a calorimeter experiment. The role of the equilibrium constant on the resulting energy release rate is similar to the biological interaction restrictions that determine how fast and to what extent two biological species react with one another. This reaction constraint has direct implications on the flow of entropy and entropy production in these systems. Consider a concentration of compound [X], which reacts with a concentration of [Y], where both are in the same buffer of [B] and produce a concentration [XY], the product. A similar development can be found in [3]
(16)$$\left[X\right]+\left[Y\right]+\left[B\right]\;\overset{yields}{\to }\;\left[XY\right]+\left[B\right]+\left[X_{F}\right]+\left[Y_{F}\right]$$

The reaction summarized in Equation (16) is constrained by the equilibrium constant, *k_B_* and the products are as shown in Equation (17), which assumes a complete reaction.
(17)$$k_{B}=\frac{\left[XY\right]}{\left[X\right]\left[Y\right]}$$

The correct form of Equation (16) that includes the equilibrium composition of the products is:(18)$${\left[X\right]}_{1}+{\left[Y\right]}_{1}+\left[B\right]\overset{yields}{\to }{\left[XY\right]}_{2}+{\left[X\right]}_{2}+{\left[Y\right]}_{2}+\left[B\right]$$

The energy and entropy balances for the described reaction can be written using absolute thermodynamic properties and are simar to that of Equations (8), (9) and (13). The transient energy balance, Equation (8), with *P_SAM_* = 0 is:(19)$$q_{g2}=\frac{d\left(m_{sam}c_{s}\left(T\right)\right)}{dt}+q_{L2}$$
where
(20)$$q_{L2}=\left(u_{HT}A\right)\left(T-T_{\infty }\right)$$

In this experiment, the compound [*X*] is injected into the sample cell that contains a fixed amount of [*Y*] = [*Y_TOT_*]. During this injection, the rate of reaction is considered to be very fast and the rate of energy released per volume, *q*_*g*2_, is not a constant and is determined from the equilibrium constant *k_B_* and the enthalpy of reaction per mol of the injected compound, Δ*H*. The rate of energy release between two specified times is:(21)$$Q_{g2}=Energy\;released\;per\;volume\;between\;t_{i+1}\;and\;t_{i}=\left([XY{]}_{i+1}-[XY{]}_{i}\right)\Delta H\;$$

Differentiating Equation (21) with respect to time determines the rate of energy released per volume:(22)$$q_{g2}=\frac{d\left[XY\right]}{dt}\Delta H$$

To determine the derivative in Equation (22), the equilibrium constant, Equation (17), is used to obtain a quadratic expression for the product [*XY*] by combining it with the species/mass balance of the species:(23)$${\left[X\right]}_{i}={\left[X\right]}_{TOT}-{\left[XY\right]}_{i}$$
(24)$${\left[Y\right]}_{i}={\left[Y\right]}_{TOT}-{\left[XY\right]}_{i}$$
(25)$${\left[X\right]}_{TOT}={\left[X\right]}_{i-1}+{\dot{X}}_{inj}t_{}$$
where the molar rate of injection is ${\dot{X}}_{inj}$ and *t* is the time from the start of the injection. The resulting relationship is
(26)$$\frac{d\left[XY\right]}{dt}=\frac{1}{2}\left\{1-\frac{\left[\left(Y_{TOT}+{\dot{X}}_{inj}t+\frac{1}{k_{B}}\right)-2Y_{TOT}\right]\;}{\left[\left(Y_{TOT}+{\dot{X}}_{inj}t+\frac{1}{k_{B}}\right)-4Y_{TOT}{\dot{X}}_{inj}t\right]}\right\}$$

It is observed from Equation (26) that the energy release rate for the reacting compound example is time dependent, unlike that in case 1. Equations (26) and (22) are used to determine the rate of energy release during the injection of compound [*X*] into the chamber filled with compound [*Y*]. The initial concentration of compound [*Y*], *Y_TOT_*, is known at the start and the injection rate of compound [*X*] is held constant. Unlike case 1, the rate of energy added to the chamber is not constant but is a function of time that starts at a maximum value and then decreases to zero, or a constant value as the number of reaction sites or available food goes towards zero, Figure 7. 

The shape of this curve reflects the trends discussed in [3], which is shown inverted since it is measuring the change in power in the two-cell calorimeter. Numerically solving Equation (19) using the energy release rate given by Equation (22) predicts the temperature response shown in Figure 8. The temperature response corresponds to the observed property in the single-cell calorimeter.

The results in Figure 8 are expected: a rise in temperature to the maximum value due to the thermal capacitance of the system, while the rate of energy release is greater than the heat transfer rate. Once the heat transfer rate overcomes the rate of energy, the temperature decreases in the system. The system returns to the temperature of the surroundings at the end of the process as expected. In terms of determining the entropy flows in this case, the observed temperature is shown in Figure 8: the heat transfer characteristics of the cell, and the thermodynamic properties of the buffer and each reactant are needed. Due to the reaction between the reactants, the absolute entropy for the products is unknown for a test case and, unlike case 1, the case of a single, non-reacting material, there are no equations of state to determine the entropy property change from the calorimeter experiment. What can be determined is the entropy flow from the chamber, since the rate of heat flow is determined in the simulation, Equation (21) (Figure 9).

Since the temperature at the system boundary is known as a function of time the rate of entropy flow from the cell can be calculated by Equation (27) and is summarized in Figure 10.
(27)$${\dot{S}}_{out}=rate\;of\;entropy\;flow\;out=\frac{q_{g2}}{T}$$

From the entropy balance on the test cell:(28)$$\dot{\sigma }=\frac{dS_{prop}}{dt}+{\dot{S}}_{out}$$
where
(29)$$\;\frac{dS_{prop}}{dt}=\Delta S_{form}^{{}^{\circ}}+\Delta S_{prop}+\Delta S_{reactants}$$

The rate of entropy flow out of the system can be determined from the observed experimental parameters. In this case, the rate of change in the entropy property cannot be determined, unless this is a known reaction and the absolute entropy values of the reactants and products are known, Equations (28) and (29). This is not usually the case when testing reacting compounds in a calorimeter because that is the purpose of the experiments. As a result, the only statement that can be made is the calculated value of the confusing parameters of the rate of entropy property change and the entropy production rate. This calculation does not allow the irreversibility of the reaction or interactions,$\;\dot{\sigma }$, in the biological communities to be determined for possible use in better understanding the dynamics of the community, the efficient use of food sources or evolutionary trends. 

In 1999, Battley [12] developed an empirical statistic model to calculate the standard molar entropy $\Delta S_{form}^{{}^{\circ}}$ with accuracy at 2%. However, the method to calculate the information of $\Delta S_{form}^{{}^{\circ}}$ from the experiment is still limited. The Gibbs free energy is defined as the maximum amount of non-expansion work in a thermodynamic system and is related to enthalpy, temperature, and entropy. The definition equation can be written as Equation (29) at thermodynamic standard states for the absolute entropy property changes [8].

In the case of drug and protein reactions, there is an additional procedural step that can be used to determine the equilibrium constant, which can then be related to the change in the Gibbs energy from which the rate of entropy property change can be calculated. This step is best illustrated from the ITC version of the calorimeter, in which a series of titrations are performed and the energy released at each step is determined as a function of the mole fraction of the product [3]. This leads to a graph that is similar to that of Figure 8 for the reacting case and the slope of the curve equals the equilibrium constant, *k_B_*. The thermodynamics of the equilibrium composition *K_B_* is related to the change in the Gibbs energy.
(30)$$\Delta G=-RTln\left[{\frac{1}{k}}_{B}\right]=\Delta G^{{}^{\circ}}=\Delta H_{f}^{{}^{\circ}}-T\Delta S_{}^{{}^{\circ}}$$

For the case of the single injection experiment described in this section, a similar curve as that developed in the ITC experiment can be determined by calculating the energy release rate within a finite time interval from the observed temperature–time response and the energy balance. The determined value of *k_B_* would then be used in Equation (30) to determine the change in the Gibbs Energy and the change in the absolute entropy property change. These results, together with Equation (26), would allow one to isolate the rate of entropy productions and measure the irreversibility of the reaction.

While the above procedure applies to a reacting system, it does not directly relate to the activities of biological microorganisms due to the lack of a clear definition of what would be the corresponding equilibrium constant. A similar parameter to the equilibrium constant for monotypic microorganisms interacting with their surroundings using a calorimeter-based measurement needs to be defined in order to separate the entropy property changes from the entropy production changes. Using probabilistic methods as described by [7] to identify a constant that characterizes the energy releasing interactions is one approach and is beyond the scope of this paper. From the comparison of case 1 and 2, it is demonstrated that the calorimeter measurement can identify the combined values of the entropy property changes and entropy production, but without further analysis or observed information, it cannot separate these parameters.

## 3. Discussion: What Is Measured

The two case examples and the calorimetry background provided illustrate that the parameter measured in these experiments is the heat flow from the test cell and it is related to the energy released by the reaction, the biological community activities, or the heat generation source. The energy flow from the cell can be further divided into the amount due to the irreversibility in the system, i.e., the entropy production term, and the amount due to the reaction. Procedures as described for the ITC type measurement that allow a parameter such as the equilibrium constant to be determined and then related to the entropy property change exist in the drug discovery or chemical reaction fields. As shown in case 2, the dynamics of the reaction in biological fields need to be developed for calorimetry to be used to isolate the entropy flows in the community. 

From the perspective of biochemistry, analysis of biomass-based on the empirical formula of various microorganisms is instructive to understand the microbe’s thermodynamic properties [12,13,14,15]. A method of the degree of reduction from elemental analysis using electron concepts estimitates the enthalpy of combustion of the biomass [14] and biosynthetic efficiency [15]. Through the oxidation reaction of the dry microorganism biomass equation, the enthalpy of formation can be determined [7]. In a living system, it is impossible to concisely express all the intermediate reactions within a microbe or the community. However, it is practicable to describe the typical microbe growth from an initial substance state to a final state comprised of the microorganism and other products by a growth-process equation [16,17]. When monitoring the heat exchange in a microbe growth experiment by an ITC calorimeter, the thermal power can be written as an exponential equation with the growth rate constant in the log growth phase [10]. Assuming there are only monotypic microbes and glucose injected into the sample cell, the growth-process equation can be written to represent the exponential phase in a traditional growth curve. To calculate the equilibrium constant, the concentration and mass of microorganisms are necessary. Makarieva et al. [18,19] developed a comprehensive database including the endogenous (non-growth) and growth metabolic rate of microorganisms and their corresponding microorganism mass involving 3006 species from all kingdoms. With an acknowledged mass of microorganisms, metabolic rate, enthalpy of biomass formation and the growth-process equation in a closed system, an energy balance can be constructed to estimate the equilibrium constant. Similarly, the equilibrium constant in the stationary phase can be determined by introducing the oxidation reaction of biomass representing the death of microbes. For more complicated cases, for example, the lysogeny by virus infection and symbiosis in biological evolution, statistical models are developed to predict the thermodynamic properties. [20,21]

An interesting observation is presented in a paper by Djamali et al. [1] in their Figure 4, Figure 5 and Figure 7, where the transient response of the observed heat flow from the test cell is plotted as a function of time. The observations in this paper are reported using the long-term, greater than approximately 10 h, which is constant heat flow with respect to time. The analysis approach described in case 2 for the single-cell injection experiment to determine an equilibrium constant-like parameter would fail when using this long-term data because of its uniform heat flow. What would be of more interest to determine the entropy flows from the calorimetry measurements are the early time data, in which the dynamics of the biological community are changing.

## 4. Conclusions

A background on the operation of calorimeters is provided and two case studies are investigated using a numerical simulation. The first case describes a single cell calorimeter containing a single-phase material excited by heat generation source function such as joule heating. In this case, the equations of state can be used to determine the energy released in the cell, as well as the entropy flows: entropy property changes, entropy flow to the surroundings and the rate of entropy production. In the second case, a reacting system, the numerical simulation of the calorimeter experiment provides the energy released and the entropy flow to the surroundings. The basic observation parameter, the temperature, cannot be used to separate the entropy property changes and the rate of entropy production. In the chemical reacting system, the data can be further analyzed to determine the equilibrium constant once the reaction equation and stoichiometric constants are specified. The equilibrium constant is then used to identify the entropy property changes and the entropy balance is used to determine the rate of entropy production. In biological conditions, the equivalent of the reaction equation and a definition of an equilibrium constant is not available for all systems. While an empirical relationship for the absolute entropy property change has been reported, it is not clear if it would apply to all systems being investigated. To use calorimetry measurements to identify the entropy flows in biological community activities, further work must be carried out to establish a framework similar to that of chemical reacting systems that are based on an equilibrium type parameter.

## Figures and Tables

**Figure 1 entropy-24-00561-f001:**
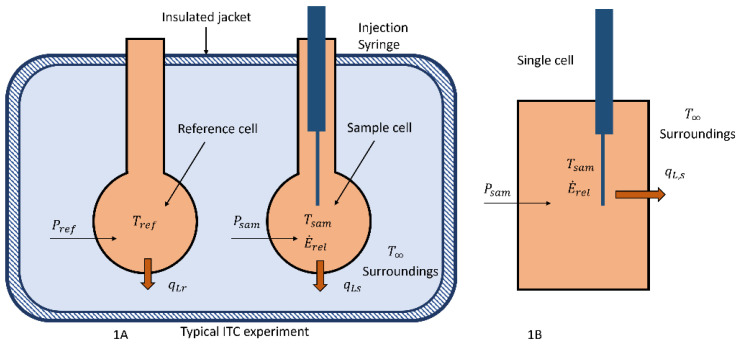
Schematic sketches of two calorimeter configurations used in determining the energy released from biological/reacting systems. (**A**) is a two-cell configuration where both cells are usually maintained at a uniform temperature and the difference between the power of the reference and sample cell is recorded. (**B**) is a single cell configuration where the temperature is monitored during the biological reaction as a function of time.

**Figure 2 entropy-24-00561-f002:**
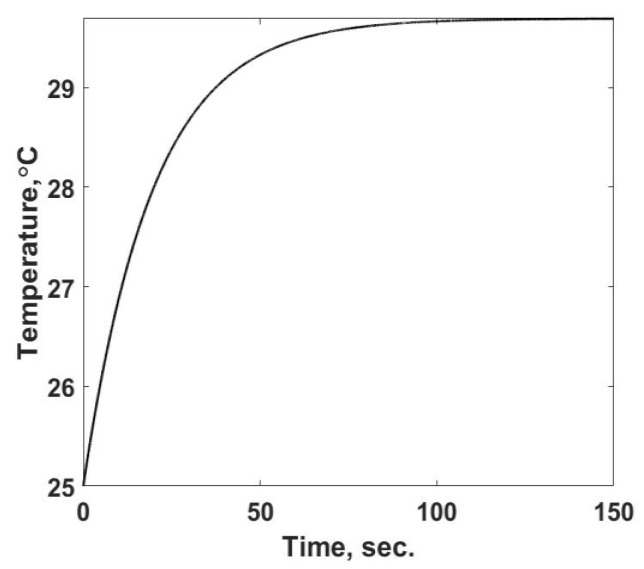
The temperature response of the single sample cell as a function of time for a pure material, water, exposed to a uniform heat generation rate.

**Figure 3 entropy-24-00561-f003:**
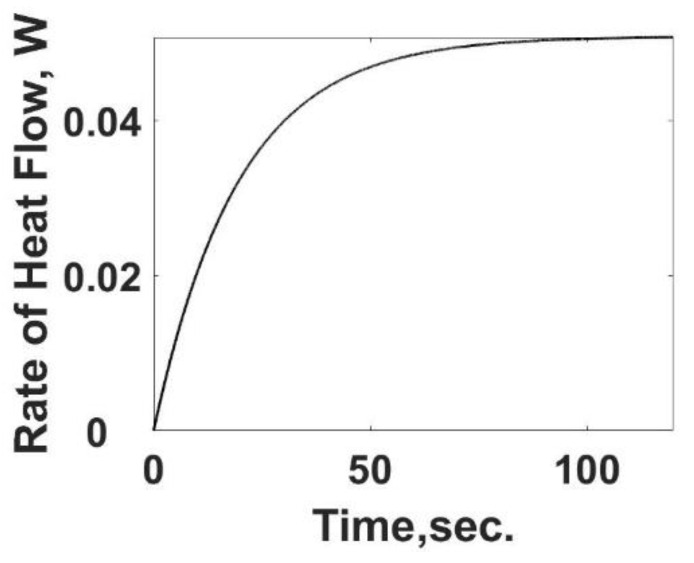
Instantaneous heat transfer rate as a function of time for the sample cell. The results shown are based on the known value of the heat transfer coefficient, (*u_HT_A*) product, and the observed temperature from Figure 2 as if it were a calorimeter experiment.

**Figure 4 entropy-24-00561-f004:**
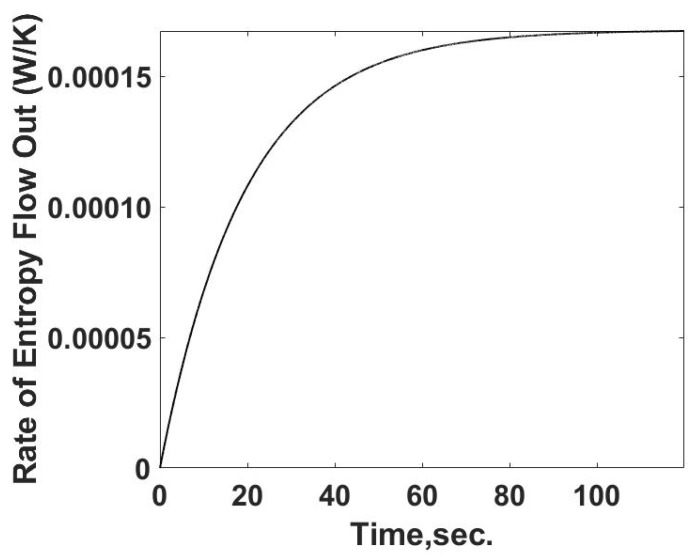
Summary of the rate of entropy flow out of the sample cell as a function of time. The calculations are based on temperatures and heat losses which would be determined from calorimetric measurements.

**Figure 5 entropy-24-00561-f005:**
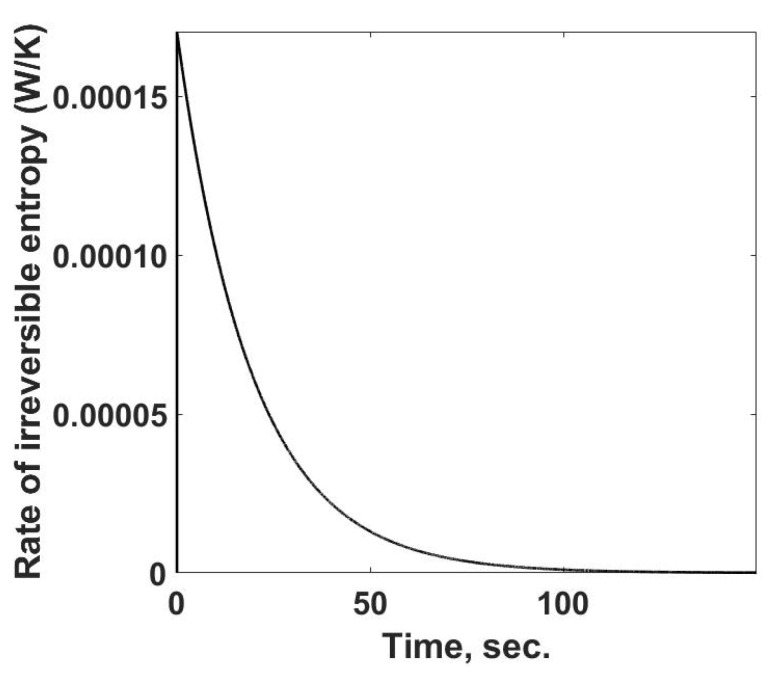
Summary of the rate of entropy property changes within the sample cell as a function of time. The calculations are based on temperatures and heat losses which would be determined from calorimetric measurements.

**Figure 6 entropy-24-00561-f006:**
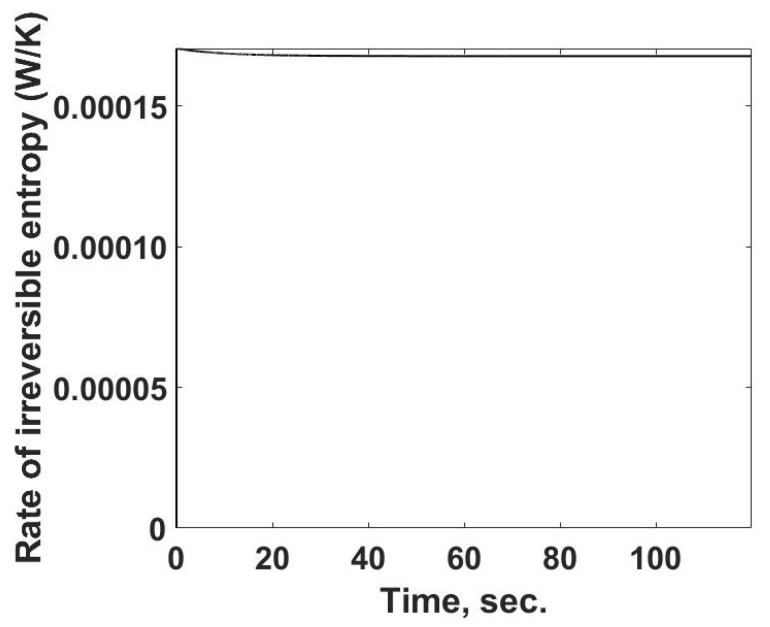
The rate of entropy production within the sample cell as a function of time. The calculations are based on temperatures and heat losses which would be determined from calorimetric measurements.

**Figure 7 entropy-24-00561-f007:**
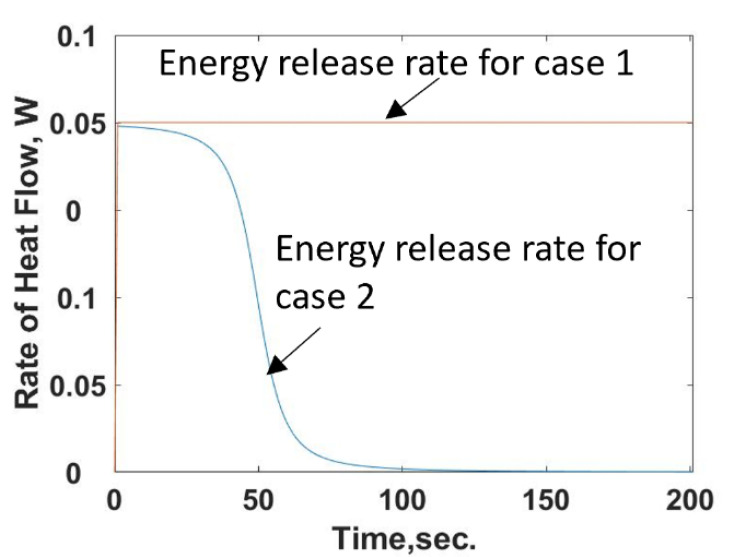
Comparison of the constant energy release rate of case 1 to the time dependent energy release rate of case 2, the reacting compound example. The orange line is the constant energy release of case 1. The blue monotonically decreasing line is for case 2.

**Figure 8 entropy-24-00561-f008:**
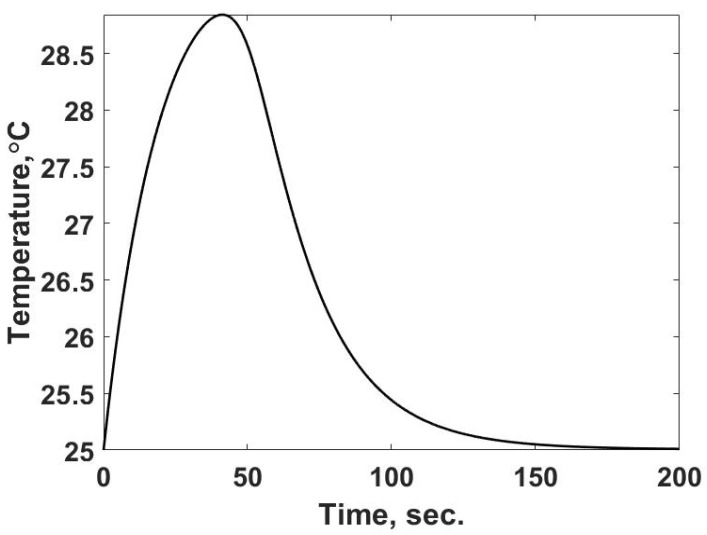
Predicted temperature vs. time response for the reacting calorimeter cell. This response corresponds to the observed parameter of the calorimeter experiment.

**Figure 9 entropy-24-00561-f009:**
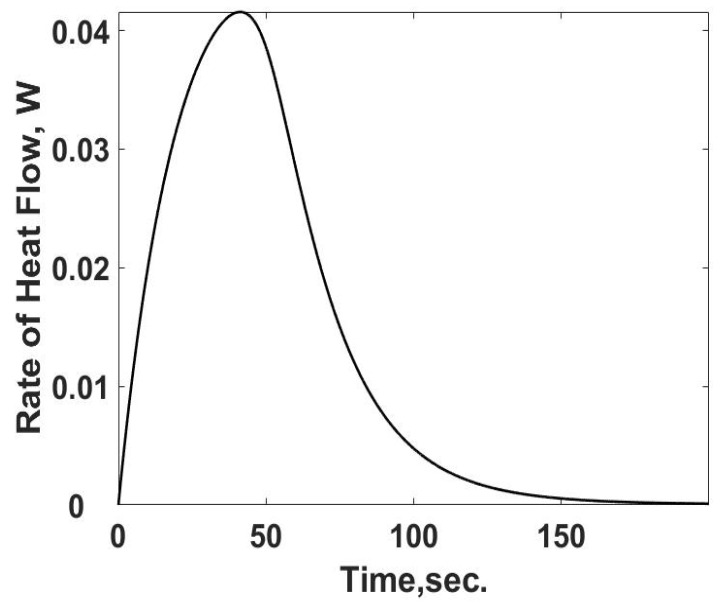
Predicted heat flow rate vs. time response for the reacting calorimeter cell. This parameter is calculated using the predicted temperature response and the heat transfer characteristics of the test cell as determined from a calibration test.

**Figure 10 entropy-24-00561-f010:**
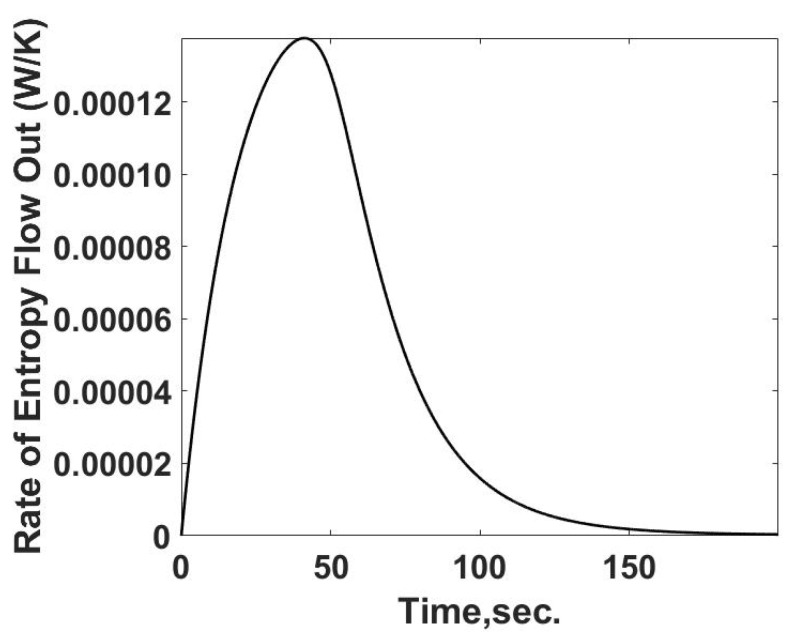
Predicted entropy flow rate out vs. time response for the reacting calorimeter cell. This parameter is calculated using the predicted temperature response and heat transfer rate.

## Data Availability

Not applicable.

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
