# Peer review of "Calorimetric Measurements of Biological Interactions and Their Relationships to Finite Time Thermodynamics Parameters"

_entropy, 2022, doi:10.3390/e24040561_

Round 1
Reviewer 1 Report
Attached file

Author Response
Our presentation which was to demonstrate that calorimetric experiments by themselves do not provide the information to isolate the entropy production term, the irreversibility, in the observed reaction. In the case of the pharmaceutical industry, calorimetry plays an important role but the additional analysis which includes an assumption of the reaction type is required to obtain the necessary entropy information. While this step is part of the commercial calorimetry package, its use toward identifying the irreversibility and potential of a drug-protein interaction is clearly understood in this process. In terms of biological processes, the role of calorimetry is increasing as is the thermodynamic and entropic interpretation of these processes. The manuscript identifies the missing steps in linking calorimetry and other known means related to growth rate formulations, degree of reduction or reactions that would allow this experiment tool to provide entropic information. This is the value of the manuscript.
We also addressed concerns of reviewer 1 regarding linking the thermodynamic framework and the biology. These edits are significant, and the comments were appreciated by the authors. The edits are identified in the file containing the track changes to the manuscript. We did not include a third test that would have included a biological reaction. The reason is that this third case would not have added to the agrument provided by the second case; without a reaction or growth model one could not separate the entropy property changes from the entropy production terms. The third case would have replaced the energy released term, the heat generation term in case 1 or the reaction constrained by the equilibrium constant, Eqs. ( 22 & 26) with an exponential power related to a growth constant characteristic of E.coli during its exponential phase. The temperature response would be a different function as would the numerical values, but the conclusion that one could not isolate the entropy production would not change. This is the focus of the paper and the foundation of future publications.
Another example of ambiguity is in lines 341 to 344, which interactions between organisms are being considered that are like equilibrium constants? Would the authors please elaborate and provide more details about the system they are investigating.
Metabolic rate in the growth curve, add a reference
Two biology communities Ty?
In a monotypic biology community, the chemical equilibrium constant is equivalent to the growth rate constant in the exponential growth phase in a traditional growth curve. (xiao) In a binary symbiosis system,
J(a,b) tangled nature model (TNM). varied case by case.
The manuscript needs to be edited for minor errors and formatting (Figures 5 and 7).
The authors should take care to differentiate between a cell in an ITC and a living cell in the text.
Changed the living cell into a living organism. Checked all other “cell” are used to describe the calorimeter.
Responding to lines 490-496: the length of calorimetry experiments with bacteria are typically on the order of hours, not seconds, reflecting the kinetics of microbial growth. Even without interference, the dynamics of a biological community may continue to change for months or years. Furthermore, the authors misuse or misunderstand the term “community”, referring to several species of organisms interacting in a common space. The term “biological interactions” also implies competition between organisms (like predation on bacteria by bacteriophage) on an ecological level but seems to be used to refer to biochemical reactions in metabolism.
Some concepts:
Biological interactions are the effect that a pair of organisms living together in a community have on each other.
A microorganism or microbe,[a] is an organism of microscopic size, which may exist in its single-celled form or as a colony of cells.
Microbial communities are groups of microorganisms that share a common living space.
The microbial community emphasizes the “shared space”. So it is not appropriate to describe two species put into one container. A similar word can choose from monotypic species, or specific bacteria and viruses.
Instead of user interaction, I changed some to activities.
Between 490-496,
Need to add some discussion about the equilibrium
The idea of using equilibrium constants in biological systems can be the metabolic rate of the bacteria or the chemical analysis from the perspective of biomass. \
Equilibrium is not at a steady state. It is dynamic.
Bring out the dissipative biology again (far from equilibrium but driven by the flow of exergy)
Line 32 (calculating)should be revised as limits in instrumentation have prevented calorimetric measurements of a single bacterium classically and technological advances are closing in on that possibility (Lee et al 2009).
I think my revision meets this requirement.
Meanwhile, from the technical point of view, there is technical limits on s
Potentials and limitations of miniaturized calorimeters for bioprocess monitoring
https://link.springer.com/article/10.1007/s00253-011-3497-7
Line 434 (eq.25) ignores the common use of calorimetry to determine the efficiency and enthalpy production of industrial-scale bioreactors. The manuscript relies heavily on a few sources and notably none of the data from cited studies is analyzed. I suggest the authors narrow down the type of biological system they want to focus on to remove the ambiguity that results from referring to biological interactions as a whole.
See Track change file
Reviewer 2 Report
Review on
"Calorimetric Measurements of Biological Interactions and
Their Relationships to Finite Time Thermodynamics Parameters"
The Authors present a heat transfer analysis related to calorimetry measurements. Although the heat transfer analysis seems quite basic, it delivers important insight to the engineering community, therefore the Reviewer supports the work. Also, as the Reviewer can see, the paper perfectly highlights the common problems in the corresponding research field.
However, before publishing the manuscript, several minor and major modifications are required due to the superficial editing. Some of the issues might be present due to the word -> pdf conversion, perhaps, but they must be corrected. Therefore the Authors are asked for major revision.
A) The equations are the part of the sentences, therefore punctuation is needed everywhere, and the text must be checked for typos. Additionally, a few sentences are recommended to reformulate since they seem inaccurate. For instance, "measure the heat dissipation on a community level", "0.5 million microbes in one-millimeter seawater", "s-1", "generate order spontaneously", etc. While these parts can be understood, it is better to be more precise and straightforward. It is also suggested to use unified notations with the same indices.
B) After Eq. (2), what is q_c? Is it the same as q_L later?
Is P_sam = P_in?
C) It is recommended to explicitly denote the time dependence of the mass in Eq. (3), and express the time derivative.
D) In row 138, it is mentioned that "the temperature of both cells will be equal". This is not clear, since one sample has different specific heat and reactions are also present.
E) In row 142, the expression "combining these two energy balances" is not clear. Eventually, (2) and (3) are the same. Or, if the Authors mean a different balance equation, please include it.
That way, Eq. (4) is also not clear and should be checked.
After that, the entire numeration system of equations is shifted by one, therefore all references throughout the text must be checked and corrected.
In row 184, it turns out that q_c is the same as q_L, indeed.
However, there is a problem with Eq. (5) (in row 215).
1) Some terms might be missing. For instance, in its present form, it seems that the specific heat depends on the temperature, but it is constant. Probably other terms are missing from the bracket in the derivative.
2) The mass is mentioned to be time-dependent. Therefore, one further equation is needed to solve this model.
Is it true?
3) q_g is a given constant with the unit of W/m^3, but the other terms are in W, therefore a volume might missing.
Additionally, Eq. (9) (in row 247) defines q_g, and now it is unclear what happens.
F) It is mentioned in row 233 that Fig. (2) shows the temperature of three spatial nodes. What are these nodes? What do you mean by "Tests at higher number of nodes confirm that the solution has converged."?
G) Based on Eq. (10), it is expected that S_out is a constant, as the same T(t) appears in the nominator and the denominator. Could you please explain Figure (4) in this respect?
H) It is suggested to reformulate the diagrams. Texts are beyond the limits, the size of the diagrams is different, and so on. Furthermore, Figure 5 partially hides a paragraph, and Figure 6 is hard to interpret. Even the curves are hardly visible and its caption is on the next page.
These shortcomings hold also for the remaining part of the manuscript.
It is also suggested to check the References as, for example, there is no Ref. [11] as it appears only by a line break in Ref. [10], and [10] is incomplete.
Overall, after thoroughly polishing the manuscript, it will be an interesting paper of high quality.
Author Response
Reviewer 2:
The Authors present a heat transfer analysis related to calorimetry measurements. Although the heat transfer analysis seems quite basic, it delivers important insight to the engineering community, therefore the Reviewer supports the work. Also, as the Reviewer can see, the paper perfectly highlights the common problems in the corresponding research field.
However, before publishing the manuscript, several minor and major modifications are required due to the superficial editing. Some of the issues might be present due to the word -> pdf conversion, perhaps, but they must be corrected. Therefore the Authors are asked for major revision.
________________________________________
A) The equations are the part of the sentences, therefore punctuation is needed everywhere, and the text must be checked for typos. Additionally, a few sentences are recommended to reformulate since they seem inaccurate. For instance, "measure the heat dissipation on a community level", "0.5 million microbes in one-millimeter seawater", "s-1", "generate order spontaneously", etc. While these parts can be understood, it is better to be more precise and straightforward. It is also suggested to use unified notations with the same indices.
Most marine microbe studies measure the heat dissipation on a community level because the isolation of individual types of microorganisms is not required. In a drop (one millimeter) of seawater, there are approximately 0.5 million microbes and 10 million viruses. To describe the distribution of individual bacteria is time-consuming and biased in their cultural practice.
Dissipative biological structures are open non-equilibrium thermodynamic systems that generate order spontaneously by directing streams of exergy
The concept of dissipative biological structure is used to describe these living systems from microorganisms to ecosystems by the dynamics of far-from-equilibrium systems.
https://www.degruyter.com/document/doi/10.1515/jnet-2018-0008/html (Ty finite)
Dissipative structures in biological systems: bistability, oscillations, spatial patterns and waves
Albert Goldbeter
https://royalsocietypublishing.org/doi/10.1098/rsta.2017.0376 (not added, just for reference)
B) After Eq. (2), what is q_c? Is it the same as q_L later?
q_L (page 8) is the heat exchange with the surrounding through convection;
Qg is the heat source
Q_c is the rate of heat transfer between the cell and the surrounding bath.
Yes, we should change ql to qc, in equations 4 & 6 are truly the same
Otherwise, we should emphasize the difference if we think they are different.
Is P_sam = P_in?
These two are different since P_sam and P_in are in different simulations/configurations a and b. Can we change the P_ref on the picture to P_in? Or change the equation from P_in to P_ref?
C) It is recommended to explicitly denote the time dependence of the mass in Eq. (3), and express the time derivative.
Insert an equation: m_SAM=m_dot_sam.
D) In row 138, it is mentioned that "the temperature of both cells will be equal". This is not clear, since one sample has different specific heat and reactions are also present.
Not act on this.
Potential explanation: One theory of the calorimeter is that the thermopile measures the reference temperature, and releases eclectic signals/currents to control the sample cell at the same temperature. The recorded electric power is an important parameter to calculate the energy change in the system.
E) In row 142, the expression "combining these two energy balances" is not clear. Eventually, (2) and (3) are the same. Or, if the Authors mean a different balance equation, please include it.
The eq.3 can be changed to U=mcdT
That way, Eq. (4) is also not clear and should be checked.
Then eq.4 can be expanded by words, like since the temperature and heat specific are constant and not changed with time, combining eq.2 and 3, and we can get..
Change the first P_SAM to qc. Not sure if I am right.
After that, the entire numeration system of equations is shifted by one, therefore all references throughout the text must be checked and corrected.
In row 184, it turns out that q_c is the same as q_L, indeed.
However, there is a problem with Eq. (5) (in row 215).
1) Some terms might be missing. For instance, in its present form, it seems that the specific heat depends on the temperature, but it is constant. Probably other terms are missing from the bracket in the derivative.
Shall we add the equation of specific heat? Or state, in this calculation, the temperature change is under 1 degree. Therefore, we are considering the specific heat to be constant under this calculation. Otherwise, the cs is changed with temperature. Since the temperature is measured, then we can get cs. However, cs is still not dependent on time. Or just be clear that this term is d(m_sam c_s T)/dt.
2) The mass is mentioned to be time-dependent. Therefore, one further equation is needed to solve this model.
Is it true? Yes, m_sam=m_dot_inj x time
3) q_g is a given constant with the unit of W/m^3, but the other terms are in W, therefore a volume might missing.
In the assumption, change q_g to q_ddot_g, and set q_g = q_ddot_g*V
Or use q_ddot_g in eq.5?
Made the change in eq.8. q_dot_L=constant =10^6 W/m^3.
Additionally, Eq. (9) (in row 247) defines q_g, and now it is unclear what happens.
Eq.9 is not a definition. It is expanded from eq.5
F) It is mentioned in row 233 that Fig. (2) shows the temperature of three spatial nodes. What are these nodes? What do you mean by "Tests at a higher number of nodes confirm that the solution has converged."?
Replot the figure from the code.
Tests at a higher number of nodes, providing higher resolution in the simulation, confirm that the solution is coverage.
G) Based on Eq. (10), it is expected that S_out is a constant, as the same T(t) appears in the nominator and the denominator. Could you please explain Figure (4) in this respect?
S_out is not a constant, since q_L varies with temperature. It should follow the trend of dT/T.
T is T(t), and it varies with time in this transient problem.
H) It is suggested to reformulate the diagrams. Texts are beyond the limits, the size of the diagrams is different, and so on. Furthermore, Figure 5 partially hides a paragraph, and Figure 6 is hard to interpret. Even the curves are hardly visible and its caption is on the next page.
Please open the PDF version. Need to replot the same size of figures.
These shortcomings hold also for the remaining part of the manuscript.
It is also suggested to check the References as, for example, there is no Ref. [11] as it appears only by a line break in Ref. [10], and [10] is incomplete.
Did not get. Ref.10 is a conference paper.
Overall, after thoroughly polishing the manuscript, it will be an interesting paper of high quality.
Round 2
Author Response
Thank you for your suggestion concerning including our comment to the editor in the introduction. We agree with you and have included it into the introduction ( see track changed version)
Below is the summary of the edits/corrections suggested by reviewer #1.

Reviewer 2 Report
The Authors are addressed all the comments, and the paper can be accepted in its present form.
Author Response
Thank you